# Homogeneity in Surgical Series: Image Reporting to Improve Evidence

**DOI:** 10.3390/jcm12041583

**Published:** 2023-02-16

**Authors:** Pietro Regazzoni, Simon Lambert, Jesse B. Jupiter, Norbert Südkamp, Wen-Chih Liu, Alberto A. Fernández Dell’Oca

**Affiliations:** 1Department of Trauma Surgery, University Hospital Basel, 4031 Basel, Switzerland; 2Department of Trauma and Orthopaedics, University College London Hospital, London NW1 2BU, UK; 3Hand and Arm Center, Department of Orthopedics, Massachusetts General Hospital, Boston, MA 02114, USA; 4Department of Orthopedics and Trauma Surgery, Medical Center, Faculty of Medicine, Albert-Ludwigs-University of Freiburg, 79106 Freiburg, Germany; 5Kaohsiung Medical University Hospital, School of Medicine, College of Medicine, Kaohsiung Medical University, Kaohsiung 80756, Taiwan; 6Department of Traumatology, The British Hospital, Montevideo 11600, Uruguay; 7Residency Program in Traumatology and Orthopedics, University of Montevideo, Montevideo 11600, Uruguay

**Keywords:** randomized controlled trials, evidence-based surgery, evidence-based medicine, ICUC, technical performance bias, image-based performance assessments

## Abstract

Good clinical practice guidelines are based on randomized controlled trials or clinical series; however, technical performance bias among surgical trials is under-assessed. The heterogeneity of technical performance within different treatment groups diminishes the level of evidence. Surgeon variability with different levels of experience—technical performance levels even after certification—influences surgical outcomes, especially in complex procedures. Technical performance quality correlates with the outcomes and costs and should be measured by image or video-photographic documentation of the surgeon’s view field during the procedures. Such consecutive, completely documented, unedited observational data—in the form of intra-operative images and a complete set of eventual radiological images—improve the surgical series’ homogeneity. Thereby, they might reflect reality and contribute towards making necessary changes for evidence-based surgery.

A recent review of the effectiveness of ten orthopedic procedures [1] noted “that most of these procedures recommended by national guidelines and used by surgeons have insufficient readily available high-quality evidence on their clinical effectiveness, which is mainly because of a lack of definitive trials.” In the absence of clinically meaningful evidence from high-quality trials, clinicians are obliged to follow the advice of the late David Sackett when discussing options for treatment with patients: “integrating individual clinical expertise with the best external clinical evidence from systematic research” [2], which often relies on consensus statements or advisory guidelines from specific institutions or professional bodies, e.g., NHS England. Evidence-Based Interventions: Guidance for CCGs [3].

A question that follows from the conclusions of this otherwise excellent article concerns whether the essential reasons for this thought-provoking conclusion have been identified, from which reliable solutions can be derived. We offer some points for debate and discussion, with a potential way forward for this challenging problem.

One obvious factor implicated, but rarely measured or assessed, in the variance within operative and non-operative treatment groups is the inter-operator variance in technical performance, whether of operative or non-operative treatment. This inevitably produces a technical performance bias (TPB) as a fundamental problem for surgical trials. The variance occurs not only during surgeons’ learning curves but also among certified professionals. The value and expectations of evidence-based medicine are undisputed [1,4,5,6,7,8]; however, for surgical trials, TPB limits the scientific adequacy of a trial and its applicability (generalizability) and acceptance [6,9,10]. Insufficient contemporaneous intraoperative performance documentation confounds a secondary analysis of the technical quality of the reported surgical procedures, as required by Item 5 of the CONSORT guidelines [7]. It is not easy to conceive how this should be achieved without documenting the technical details of the surgical procedure with still images or video clips of the operation field and all intra-operative images [11]. It is interesting that Blom et al. [1] report that total knee replacement, a procedure highly dependent on the proper use of instrumentation, is one of only two procedures of the ten studied for which there is sufficient evidence to support its use in the specific indication of end-stage osteoarthritis of the knee. By removing variability in the surgeons’ performance through instrumentation, including augmented or robotic assistance, the variance in the procedure outcome could be reduced, thus making a comparison with non-operative interventions more meaningful, measurable, and relevant (for instance, in cost-analysis comparisons of treatments). It will be interesting to speculate whether navigated (‘robotic’) knee replacement will take this further [12], making the individual surgeon’s performance even less influential for the outcome [13]. The second procedure, for which there is sufficient evidence for efficacy (carpal tunnel decompression, a procedure in which the essence of technical success is soft tissue handling, i.e., surgical competence), comprises fewer ‘steps-to-success’ to master, and variability may therefore be minimized between surgeons. The quality of the various technical aspects of surgery, such as the expertise demonstrated in soft tissue handling or the number, force, and amplitude of maneuvers needed for fracture reduction—essential for an assessment of performance and procedure outcome—are not documented in most studies and cannot, therefore, be considered. The homogeneity of the technical aspects of different treatment groups in a clinical study is indispensable in a skill-dependent field such as surgery [14,15] but is rarely reported. Current methods for documenting and selectively recording x-rays without unedited contemporaneous, e.g., a video–photographic representation of procedures, do not appear sufficient to guarantee the needed homogeneity. In addition, the complete documentation of all the surgical procedures helps to build up supervised machine-learning models. The latest artificial intelligence (AI) technology assists in the automatic post-production of the key steps of still images and short video clips for a rapid use with a high accuracy. An AI-based surgical platform has played a role in some specific endoscopically assisted procedures [16], and a similar technology may apply to other surgeries in the future.

Currently, the homogeneity of a technical performance within different treatment groups appears so sufficiently poor that the evidence level deteriorates [17]. This has inevitably occurred in frequently cited randomized controlled trials (RCTs) such as the ProFHER study [18,19] regarding the treatment of proximal humerus fractures and the UK heel fracture trial [20] and the UK DRAAFT trial regarding the treatment of distal radius fractures [21]. The conclusions of such studies lead to recommendations that may not be directly relevant to the individual patient and are therefore of limited value in clinical practice [22] Efforts are needed in surgery to produce evidence levels similar to those generated in internal medicine. Justifications for surgical decision-making, such as ‘this works in my hands’ or ‘what my mentor taught me’ [23], should be replaced by scientific evidence. Operative procedures, in particular, the experience and preferences of surgeons, which reflect the surgeons’ performance, must be stratified. The goal(s) of the treatment must be defined before the intervention, independent of the chosen treatment modality. Subsequently, a surgical outcome is influenced by preoperative expectations [24,25] and surgical performance. The post-procedure assessment of whether the goals were met in the different treatment groups is indispensable: the decrement (including complications caused by suboptimal surgical performance) after the procedure matters as least as much to patients as the increment of functionality gained. The reasons for differences (decrements) between ‘work as planned’ and ‘work as done’ must be analyzed. Goals—such as an ‘anatomical’ reconstruction of a fracture, not an approximation to it—are sometimes only reached by technically highly skilled surgeons, especially for infrequent pathologies. An unrecorded but poor performance from non-specialized surgeons with wildly different experience levels might lead to poorer outcomes and failure to attain the desired goals [26,27,28]. TPB compounds the problem of ‘group inhomogeneity’ inherent to many classifications of disease used in such trials: inconclusive results are almost inevitable.

Clinical trials reported without the contemporaneous recording of imaging data, including video–photographic documentation, permitting an independent retrospective evaluation of both group homogeneity (of the classifications used, patients’ characteristics, etc.) and the technical performance quality, lose scientific value. The technical performance quality is measurable and correlates with the outcomes and costs [14,29] in cardiac, visceral, and video-assisted surgery studies. It is difficult to imagine that such correlations should not be valid for other fields of surgery if the technical metrics are adapted. The performance–outcome effect might increase with the complexity of the procedure: discussions could then arise about what is technically straightforward and what is not and at what level of expertise a surgeon must be to accomplish a particular procedure. From one surgeon to another, a critical variability exists in soft tissue handling and the sequence of intricate actions to reach articular congruity. This produces an inevitable and undesired inhomogeneity.

The inherent heterogeneity of complex interventions [17] is well known; nevertheless, surgical RCTs seldom consider potentially different quality levels of the technical performance [30]. This is relevant to RCTs in medicine; as a doctor (surgeon), dependent factors are much more critical. Defining necessary and homogeneous performance quality factors can therefore improve the outcomes. The absence of standards of performance assessments for every surgical specialty cannot be a reason not to initiate an effort to establish them. Intra-operative procedural documentation will be needed to determine a ‘performance gap’: the difference between a high and a low level of performance of a specific technical act. Quality levels can be defined on the basis of complete intra-operative image documentation [14]. This might comprise a rating of a specific procedure step or the entire procedure; surgical time-to-completion does not necessarily reflect either expertise or accuracy but is often used as a surrogate for these performance dimensions. Such performance assessments are still to be clearly defined but all will likely be image-based [31]. To assume that a defined written protocol guarantees that all procedures follow a uniform sequence of actions according to the protocol are illusory. This is particularly true in trauma due to the essential variations from one case to another, which are difficult to depict in a classification.

In one attempt to contribute to this lack of standards, the ICUC working group [32] has developed a concept for complete and detailed image-based reporting, including unedited, contemporaneous, and complete photo-documentation of entire procedures. Such documentation has the potential to overcome the previously mentioned TPB as it allows secondary, retrospective, and independent analysis [32]. The completeness of the record allows significant help for learning by providing images of technical details. It also defines the value of the initiative: all critical or key steps and potential shortcomings are included [6,33]. The evidence-based justification of technical practices based on RCTs in (orthopedic) surgery is a laudable goal but equally challenging to realize. There are relevant reasons for this reality.

First, the standardization of the key steps of any surgical procedure is not only difficult—especially in multi-center trials—but also insufficient if no agreed metrics for secondary analysis and comparison exist. Second, technical performance bias or inhomogeneity (within study groups containing very different elements or classifications applied to such groups) are the basis of imprecise or even incorrect conclusions, which therefore ‘permit’ a reversion to less evidence-based medicine. Finally, RCT data represent ‘work as planned’ (according to a research protocol); the attainment of ‘work as planned’ (the ideal outcome) rather than ‘work as done’ (the actual outcome) is possibly only realized by a minority of surgeons, and not representative of what most surgeons do in their daily practice. Consecutive, completely documented, unedited observational data might reflect reality more precisely while fulfilling the requirements of the Cochrane Collaboration [11].

## Consequences and Conclusions

Transparent (unedited) intraoperative image data, allowing a retrospective analysis, are indispensable to avoid a technical performance bias and assure the homogeneity of treatment groups in surgical trials. Complete, continuous clinical series can represent ‘real world data’ better than RCTs if they avoid these biases. The incidence of inconclusive results, frequent in surgical RCTs, could diminish. Following the ICUC concept of a complete intra-operative image documentation of surgical procedures, we can obtain data allowing for a retrospective analysis. This would contribute to necessary changes toward evidence-based surgery (EBS).

## Data Availability

Not applicable.

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
