# Peer review of "Homogeneity in Surgical Series: Image Reporting to Improve Evidence"

_jcm, 2023, doi:10.3390/jcm12041583_

Round 1
Reviewer 1 Report
Dear editor,
Dear authors,
Thank you very much for the opportunity to review this viewpoint paper.
Generally, the authors discuss the dependence of surgical RCTs and studies on the technical performance and on the surgeons’ skills. The discuss a way to avoid the bias related to these both facts.
The viewpoint is well written and interesting and presents an effort improving evidence-based surgery.
But so far, the reviewer would like to include the following:
1. At which point of a RCT and to which extent the authors assume a higher effect of technical and surgeon dependent factors than of patient specific parameters.
2. Which factor is affected: objectivity, reliability, and validity? How plan the authors to control these by their planned way to avoid technical and surgeon dependent changes of study results?
3. To which extent the planned way would improve or deteriorate the surgical training and education?
4. Which instruments the authors aimed to use to assess objectively the technical and surgical performance?
Author Response
Thank you very much for the opportunity to review this viewpoint paper. Generally, the authors discuss the dependence of surgical RCTs and studies on the technical performance and on the surgeons’ skills. They discuss a way to avoid the bias related to these both facts. The viewpoint is well-written and interesting and presents an effort to improve evidence-based surgery.
But so far, the reviewer would like to include the following:
- At which point of an RCT and to which extent the authors assume a higher effect of technical and surgeon dependent factors than of patient specific parameters.
Thanks for the comments. We agree with your comments that RCTs are based on trials by specialists. A surgical RCT compares several surgical procedures in a specific group of patients, e.g., the WRIST trial. A secondary analysis study, based on the data for the complete database, sometimes tries to figure out the correlation between surgical complications with some patient-specific parameters. This analysis is based on the hypothesis that all surgical procedures follow a uniform sequence of actions according to the protocol; however, it is only possible to audit all surgical procedures with complete surgical images.
- Which factor is affected: objectivity, reliability, and validity? How plan the authors to control these by their planned way to avoid technical and surgeon dependent changes of study results?
Thank you for the comments. On the fact that only a complete perioperative image can allow a supervisor or an independent reviewer to audit the surgical results more objectively, and reliably in a valid approach.
- To which extent the planned way would improve or deteriorate the surgical training and education?
Thank you for the comments. We considered gathering a set of complete image-based surgical materials for surgeons with a lack of experience to learn from.
- Which instruments the authors aimed to use to assess objectively the technical and surgical performance?
Thank you for the comments. The surgical fellowship is an apprenticeship, and a fellow surgeon builds up surgical skills under the supervision of a mentor. There is a variety of forms to audit the surgical performance of a surgeon under training. When a surgeon work independently, it is difficult to audit the surgical performance without complete intraoperative surgical records. Nowadays it is possible to capture still images and short video clips during operation and evaluate it postoperatively. With the advance of AI technology built on supervised machine learning models in a large dataset of complete intraoperative media information, the assessment of surgical performance will become more objective and efficient.

Reviewer 2 Report
Title:Ok.
Abstract: consider adding to the abstract the purpose of this study.
-What about the balance between different studies? In most parts of the world, surgeries are typically performed by specialists. For routine surgeries performed in large numbers, it can be assumed that every surgeon has surpassed the learning curve. On the other hand, studies that focus on rarer surgeries tend to be case series, and there is less emphasis on meta-analyses or randomized controlled trials (RCTs) due to the small patient population.
-The article discusses a method for processing data by analyzing images from intraoperative fluoroscopy. In my opinion, this approach is only applicable to certain surgical procedures that require intraoperative fluoroscopy, but there are many operations that do not use this technology. How can these surgeries be addressed?
Additionally, In the article, it is possible to mention existing technology, in arthroscopic surgeries, (first used in laparoscopes), in a recording of the entire surgery is made and then edited down to key points. Although this is currently not used for research, tools do exist, and technology is advancing towards the development of innovative and relevant teaching tools.
https://theator.io/
-In your opinion, will the provision of balanced and informative data, similar to that in internal medicine, result in an improvement or change in outcomes? Can you explain how improved reporting and scientific writing can truly lead to better results? In my view, there needs to be a strong emphasis on the need to enhance research methods, and a direct impact on medical education and training. Please elaborate on this and provide additional insights on the topic of research.
Line 88 “Subsequently, a post-procedure assessment of whether the goals were met or not in the different treatment groups is indispensable: the decrement (including complications caused by suboptimal surgical performance) after the procedure matters as least as much to patients as the increment of functionality gained."
Please consider adding or elaborating this paragraph further with regards to patients' expectations
Chahla J, Beck EC, Nwachukwu BU, Alter T, Harris JD, Nho SJ (2019) Is There an 238
Association Between Preoperative Expectations and Patient-Reported Outcome After 239
Hip Arthroscopy for Femoroacetabular Impingement Syndrome? Arthroscopy 240
35:3250-3258.e1
Factor S, Neuman Y, Vidra M, Shalom M, Lichtenstein A, Amar E, Rath E. Violation of expectations is correlated with satisfaction following hip arthroscopy. Knee Surg Sports Traumatol Arthrosc. 2022 Oct 1. doi: 10.1007/s00167-022-07182-1. Epub ahead of print. PMID: 36181523.
Author Response
- Title: Ok.
Thank you.
- Abstract: consider adding to the abstract the purpose of this study. By completing the intraoperative imaging.
Thank you for the kind reminder. The study aims to demonstrate consecutive, completely documented, unedited observational data – in form of intra-operative images and a complete set of eventual radiological images – to allow for improving the homogeneity of surgical series. We added the paragraph on Lines 28-30.
- What about the balance between different studies? In most parts of the world, surgeries are typically performed by specialists. For routine surgeries performed in large numbers, it can be assumed that every surgeon has surpassed the learning curve. On the other hand, studies that focus on rarer surgeries tend to be case series, and there is less emphasis on meta-analyses or randomized controlled trials (RCTs) due to the small patient population.
Thanks for the comments. We agree with your comments that RCTs are based on trials by specialists. A surgical RCT compares several surgical procedures in a specific group of patients, e.g., the WRIST trial. A secondary analysis study, based on the data for the complete database, sometimes tries to figure out the correlation between surgical complications with some patient-specific parameters. This analysis is based on the hypothesis that all surgical procedures follow a uniform sequence of actions according to the protocol; however, it is only possible to audit all surgical procedures with complete surgical images.
- The article discusses a method for processing data by analyzing images from intraoperative fluoroscopy. In my opinion, this approach is only applicable to certain surgical procedures that require intraoperative fluoroscopy, but there are many operations that do not use this technology. How can these surgeries be addressed?
Thank you for the observations. The title of this manuscript is “Homogeneity in Surgical Series: Image-reporting to Improve Evidence.” We want to mention the importance of intra-operative images, not limited to intraoperative fluoroscopy. We re-wrote the paragraph in Abstract and Lines 57-60.
- Additionally, In the article, it is possible to mention existing technology, in arthroscopic surgeries, (first used in laparoscopes), in a recording of the entire surgery is made and then edited down to key points. Although this is currently not used for research, tools do exist, and technology is advancing toward the development of innovative and relevant teaching tools.https://theator.io/
Thank you for the suggestions. We added a paragraph in Lines 81-86 describing this existing latest AI technology to strengthen our viewpoint that complete documentation of all the surgical procedures helps to build up supervised machine-learning models and a new AI-based surgical platform may play a role in some specific endoscopically assisted procedures.
- In your opinion, will the provision of balanced and informative data, similar to that in internal medicine, result in an improvement or change in outcomes? Can you explain how improved reporting and scientific writing can truly lead to better results? In my view, there needs to be a strong emphasis on the need to enhance research methods, and a direct impact on medical education and training. Please elaborate on this and provide additional insights on the topic of research.
Thank you for the comments. We added a paragraph “The completeness of the record allows significant help for learning by providing images of technical details. It also defines the value of the initiative: all critical or key steps and potential shortcomings are included” in Lines 151-152.
- Line 88 “Subsequently, a post-procedure assessment of whether the goals were met or not in the different treatment groups is indispensable: the decrement (including complications caused by suboptimal surgical performance) after the procedure matters as least as much to patients as the increment of functionality gained." Please consider adding or elaborating this paragraph further with regard to patients' expectations.
Thank you for the comments. We agreed with you that both preoperative expectations and surgical performance influenced surgical outcomes. However, the study focuses on the importance of surgical performance through an image-based post-procedure assessment. We re-wrote the paragraph on Lines 100-102.

Round 2
Reviewer 2 Report
This study entitled “Homogeneity in Surgical Series: Image-reporting to Improve Evidence” is overall well written. The authors addressed my previous comments.